# Two Single Nucleotide Polymorphisms in the Von Hippel-Lindau Tumor Suppressor Gene in Patients with Clear Cell Renal Cell Carcinoma

**DOI:** 10.3390/ijms24043778

**Published:** 2023-02-14

**Authors:** Magdalena Chrabańska, Nikola Szweda-Gandor, Bogna Drozdzowska

**Affiliations:** 1Department and Chair of Pathomorphology, Faculty of Medical Sciences in Zabrze, Medical University of Silesia, 40-055 Katowice, Poland; 2Department and Clinic of Internal Medicine, Diabetology and Nephrology, Medical University of Silesia, 41-800 Zabrze, Poland

**Keywords:** renal cell carcinoma, single nucleotide polymorphisms, VHL

## Abstract

The most common subtype of renal cell carcinoma (RCC) is clear cell type (ccRCC), which accounts for approximately 75% of cases. von Hippel-Lindau (*VHL*) gene has been shown to be affected in more than half of ccRCC cases. Two single nucleotide polymorphisms (SNPs) located in *VHL* gene, rs779805 and rs1642742, are reported to be involved in the occurrence of ccRCC. The aim of this study was to assess their associations with clinicopathologic and immunohistochemical parameters, as well as risk and survival of ccRCC. The study population consisted of 129 patients. No significant differences in genotype or allele frequencies of *VHL* gene polymorphisms were observed between ccRCC cases and control population, and we have found that our results do not indicate a significant relationship of these SNPs with respect to ccRCC susceptibility. Additionally, we did not observe a significant association of these two SNPs with ccRCC survival. However, our results conclude that rs1642742 and rs779805 in the *VHL* gene are associated with increased tumor size, which is the most important prognostic indicator of renal cancer. Moreover, our analysis showed that patients with genotype AA of rs1642742 have a trend towards higher likelihood of developing ccRCC within their lifetime, while allele G of rs779805 can have a preventive effect against the development of renal cancer in stage 1. Therefore, these SNPs in *VHL* may be useful as genetic tumor markers for the molecular diagnostics for ccRCC patients.

## 1. Introduction

Renal cell carcinoma (RCC) is the predominant form of malignancy of the kidney and accounts for about 85% of all renal cancers [1,2,3]. The most common histological subtype of RCC is clear cell renal cell carcinoma (ccRCC) [2,4]. Despite recent advances in the diagnosis and treatment of RCC, it remains a tumor of unpredictable presentation and clinical outcome. The clinical outcomes of RCC differ widely, indicating the need for appropriate and accurate prognostic parameters. At present, prediction of RCC prognosis largely depends on the conventional prognostic factors such as histological features, nuclear grade, and TNM staging system [1,2,5,6]. In addition to clinicopathological prognostic features, several immunohistochemical markers, such as CD44, MMP-2, and MMP-9, have been investigated as potential prognosticators for ccRCC [7,8,9]. CD44 is a multifunctional class I transmembrane protein which binds to the extracellular matrix (primarily to hyaluronic acid). It is involved in the cancer cell aggregation, migration, activation and metastasis. CD44 has recently been introduced as a cancer stem cell marker in various types of cancers, including renal cell carcinoma. Overexpression of CD44 has been linked to tumor progression, recurrence, metastasis and resistance to the chemotherapeutic agents [7,8], whereas matrix metalloproteinases (MMPs) are a family of zinc-dependent endopeptidases that are collectively capable of degrading most components of the basement membrane and extracellular matrix, facilitating cell migration. In particular, the ability to degrade type IV collagen, the major component of the basement membrane, is unique to MMP-2 and MMP-9. A significant correlation between increased MMPs and poor prognosis, including shortened patient survival, has been documented in many neoplasms, including renal cancer [9]. However, these parameters are not enough to significantly enhance patient management strategies. Therefore, in light of novel therapy choices, other novel prognostic parameters including genetic variations are needed to accurately predict the survival of ccRCC patients. Recently, some studies showed that genetic polymorphisms of candidate genes are associated with susceptibility and/or prognosis in patients with RCC [6,10,11,12,13]. Considering the important role of von Hippel–Lindau (VHL)/hypoxia-inducible factor 1 subunit alpha (HIF1α) pathway in stimulating angiogenesis, inducing tumor cell proliferation, invasion, and metastasis in ccRCC, it is possible that polymorphic variations in the Von Hippel-Lindau gene (*VHL*) may have an influence on the susceptibility or prediction of ccRCC [6]. The *VHL* tumor suppressor gene is located on the short arm of chromosome 3 mapped to 3p25-3p26 and comprises 639 nucleotides in 3 exons encoding 213 amino acids [14,15,16]. The VHL gene product (pVHL) has been investigated as a multi-adaptor protein, interacting with many different binding partners. Its best described function is to target other proteins for ubiquitination and proteasomal degradation as a component of an E3 ubiquitin protein ligase. Among its targets are the HIF subunits 1α and 2α (HIF1α and HIF2α), which upregulate many genes important in metastatic processes [17,18,19,20,21,22]. Under standard oxygen conditions, the VHL protein forms a complex with elongin B, elongin C, and cullin 2 which targets hydroxylated HIF1α for ubiquitin-mediated degradation. Under hypoxic conditions, the VHL complex cannot bind HIF1α for degradation. Therefore, HIF1α accumulates, which results in transcription of additional genes that facilitate oxygen delivery, cellular adaptation to oxygen dedecline, and angiogenesis [23,24,25]. In addition to destabilizing effect of pVHL on HIF1/2α, it is also involved in the recruitment of many effector proteins to regulate a variety of cellular processes including microtubule stability, activation of p53, neuronal apoptosis, cellular senescence and aneuploidy, ubiquitination of RNA polymerase II and regulation of NFkB activity [26]. More than half of ccRCC cases are associated with either *VHL* gene mutations or transcriptional repression with promoter and first exon regions hypermethylation [13,14]. In existing literature, two single nucleotide polymorphisms (SNPs), rs779805 and rs1642742 involving both A and G, located in the promoter and 3′ untranslated regions of the *VHL* gene are reported to be involved in the occurrence of ccRCC [6,14,26,27,28].

The aim of this study was to clarify the influence of these SNPs in *VHL* on ccRCC risk and survival. Moreover, we assessed their associations with clinicopathologic parameters and immunohistochemical expression of CD44, MMP-2, and MMP-9 in ccRCC cases.

## 2. Results

### 2.1. General Characteristics of Patients

A total of 129 patients with ccRCC were included in the study. The mean age of the patients was 63 years (SD = 10.7), ranging from 34 to 85 years. Tumor size ranged from 1 to 18 cm at the largest diameter. The mean duration of follow-up was 71.7 months (SD = 11.4), with a median of 72.7 months. The clinicopathological characteristics of patients are provided in Table 1.

### 2.2. Immunohistochemical Staining

A total of 129 ccRCC cases were analyzed immunohistochemically for CD44, MMP-2, and MMP-9 expression in the neoplastic cells. In the case of CD44 expression, 58 (44.9%) samples were negative, while upregulation of CD44 molecule was found in 71 (55.1%) cases. Of all samples stained, 121 (93.8%) and 118 (91.7%) did not show any staining for MMP-2 and MMP-9, respectively. Distributions of CD44, MMP-2, and MMP-9 expression based on overall immunohistochemical score are presented in Table 2.

### 2.3. Genotype and Allele Frequencies of VHL Polymorphisms in the ccRCC and Control European Population

Allele frequencies and genotype distributions of *VHL* polymorphisms in patients and control populations are shown in Table 3. All information about control population came from the Ensembl database https://www.ensembl.org/index.html (accessed on 5 January 2023). The control population consisted of the European (northern and western) population. All observed genotype frequencies in both controls and patients conform to the Hardy–Weinberg equilibrium. Comparing the study and control groups, our analysis showed a shift in the genotype frequency values for rs1642742. It may be related to either the different size of the subgroups or may be associated with clinical context. In the future, we will try to expand the research to explain these discrepancies in genotype frequency values. Such an investigation may result in a very interesting prognostic study for RCC patients and their families.

### 2.4. Effects of VHL Polymorphisms on Clinicopathological Characteristics of ccRCC Patients

Borderline statistically significant *p*-values were observed when assessing the association between the evaluated SNPs (rs1642742 and rs779805 in the *VHL* tumor suppressor gene) and increased tumor size. The associations of *VHL* polymorphisms with tumor diameter are shown in Table 4.

Table 5 shows the associations of VHL polymorphisms with pathologic tumor (T) stage of ccRCC. Our analysis showed that patients with genotype AA of rs1642742 have a trend towards higher likelihood of developing ccRCC within their lifetime, while allele G of rs779805 can have a preventive effect against the development of renal cancer in stage 1.

There were no significant associations between the evaluated SNPs and patient gender, patient age, tumor grade, presence of tumor necrosis, presence of sarcomatoid and rhabdoid differentiation, presence of fibrous renal capsule invasion, presence of perinephric fat invasion, presence of renal sinus fat invasion, or presence of vascular invasion of renal sinus vessels.

### 2.5. Associations of the VHL Polymorphisms with CD44, MMP-2 and MMP-9 Expression in ccRCC Tumors

We found no significant associations between the evaluated SNPs and CD44, MMP-2 and MMP-9 expression in ccRCC tumors. The associations of *VHL* polymorphisms with CD44, MMP-2 and MMP-9 expression levels are shown in Table 6.

### 2.6. Effects of VHL Polymorphisms on ccRCC Overall Survival

Statistical analysis was used to assess the associations between genotypes of the two SNPs in *VHL* and survival of patients with ccRCC. As shown in Table 7, we did not observe a significant association of each allele and genotype of these two SNPs with survival of ccRCC.

## 3. Discussion

Renal cell carcinoma has the highest mortality rate of the genitourinary cancers and the incidence of this neoplasm has risen continuously. If detected early, the cancer is curable by surgical excision of the tumor, although a minority are at risk of recurrence. RCC is a heterogeneous group of neoplasms and comprises several histological subtypes with different genetics, biology and behavior. The most common histological type is ccRCC, which represents 75–80% of all RCC [29]. The etiology of most cases of ccRCC is still not fully understood. Cigarette smoking, obesity, hypertension and/or related medications have been implicated as risk factors, although the increase in risk is relatively inconspicuous [30]. More than 96% of ccRCC are sporadic tumors [31]. Chromosome 3p deletion and inactivation of the *VHL* suppressor gene is the most common genetic alteration. Inactivation of *VHL* is specific to ccRCC and is not found in other histological subtypes of RCC. Some would argue that alteration of *VHL* is the initiating event in most sporadic cases of this neoplasm, because *VHL* inactivation is so common in sporadic ccRCC and, in cases where ccRCC shows only one chromosomal loss, it is consistently 3p [29]. Approximately 66% of sporadic ccRCC have biallelic inactivation of *VHL*, and about 10–15% have promoter hypermethylation associated with transcriptional inactivation. Loss of heterozygosity (LOH) of 3p including the *VHL* gene locus is present in 87% of ccRCC, while homozygous deletion and rearrangement account for further cases of biallelic inactivation [32,33]. The protein encoded by the *VHL* gene is a component of the elongin complex and is involved in the ubiquitination and degradation of hypoxia-inducible-factor (HIF), which is a transcription factor that plays a main role in the regulation of gene expression by oxygen. *VHL* inactivation in ccRCC leads to the activation of the hypoxia pathway via HIF1α and HIF2α, which sequentially activates expression of genes involved in the hypoxia response and angiogenesis [33,34].

Two SNPs, rs779805 and rs1642742 involving both A and G and located in the *VHL* gene, are informative and implicated in the occurrence of ccRCC [6,14,35,36,37]. Therefore, in the present study, we investigated the associations of these polymorphisms with risk, clinicopathological characteristics, CD44, MMP-2, and MMP-9 immunoreactivity and survival of ccRCC. We could not find any significant difference in the *VHL* genotype and allele frequency between ccRCC patients and a control population, which may suggest that these *VHL* polymorphisms are not associated with the onset of RCC, however further studies with a larger sample size are needed to clarify this phenomenon. These results are consistent with those obtained by Qin et al. [6], who studied polymorphic variations in *HIF1A* and *VHL* (including rs779805) in 518 cases of ccRCC and by Bensouilah et al. [38] who investigated the role of fourteen SNPs in the different genes including rs779805 and rs1642742 in *VHL* in 54 patients with ccRCC. On the other hand, Wang et al. [14] found that rs1642742 is a more sensitive risk factor for sporadic RCC than rs779805, however they investigated these two SNPs in only 19 cases of RCC, and such a small sample size may not be enough to make broad generalizations. It should also be noted that studies cited above [6,14,38] were conducted on East Asian and North African populations, which could account for the differences compared to our study conducted on a Central European population. Moreover, Wang et al. [39] reported that the G allelic frequencies in both rs779805 and rs1642742 of *VHL* gene in healthy subjects from Taiwan are much lower than in the European population. Thus, the differences between different populations may also apply to the ccRCC cases.

In some research [14,39] a positive correlation between RCC and G variant has been indicated, suggesting that GG homozygote is a lethal genetic mutation causing early cell death before clinical cancer can develop, and results in the lack of GG homozygote in the RCC patients in Asian population. This means that, if the G allele carriage at rs779805 and rs1642742 or AG heterozygote frequency increases, susceptibility to RCC increases. However, further studies with a larger sample size are needed to verify this hypothesis.

Some authors have tried to assess the clinicopathological characteristics in relation to polymorphic variations in *VHL* gene in patients with RCC. On multivariate analysis, Wang et al. [14] found that age and gender were factors in the genotype distributions of both rs779805 and rs1642742, however adjustment for tumor stage did not significantly affect the genotype distributions. Qin et al. [6] studied *VHL* polymorphic variations in rs779805 in *VHL* and in 3 SNPs in *HIF1A* (rs11549465, rs11549467, and rs2057482) and observed no significant associations between the evaluated genotypes in *VHL* gene and primary tumor stage, distant metastasis, tumor size or tumor grade. However, in combined analysis, patients with ≥2 variant alleles of these four polymorphisms were significantly less likely to present with lymph node metastasis, and significantly more likely to present with a localized clinical stage. On the contrary, borderline *p*-values (*p* = 0.048 and *p* = 0.05) were observed in our study when assessing the association between the evaluated SNPs and increased tumor size.

Several studies have reported that *VHL* alternations may have effects on the prognosis of RCC with inconclusive results. However, it has been suggested that the most useful role for *VHL* mutational status may be in combination with other potential biomarkers, such as *VEGF* polymorphisms [40,41,42,43,44]. Qin et al. [6] also investigated the associations between the genotypes of rs779805 in *VHL* and survival of patients with renal cancer, but they observed no significant associations between the evaluated SNP and patients’ survival. However, they found that variant alleles (≥1 vs. 0) of the four polymorphisms in *VHL* and *HIF1A* genes were an independent risk factor for RCC survival.

Our study has some limitations. First, it was conducted on relatively small group of patients and may not have enough statistical power to show significant associations. A subsequent limitation concerned the limited amount of prior research on this topic. Available research was performed on different ethnic groups and, in some cases, on a very small group of subjects, which may result in some discrepancies. Overall, we consider this research to be relatively preliminary and plan further studies with larger number of patients.

## 4. Materials and Methods

The standard methodology of this study was reported according to “Strengthening the Reporting of Observational Studies in Epidemiology” (STROBE) guidelines [45]. The protocol of this study was approved by the Institutional Review Board of Medical University of Silesia, Katowice, Poland (PCN/CBN/0052/KB/243/22). Patient data were kept fully anonymous in all steps.

### 4.1. Patient Characteristics and Tumor Samples

The population study consisted of 129 patients from Upper Silesia in Poland with histologically confirmed ccRCC. Cases with other histological RCC subtypes were not included in this study. All tumor specimens had been obtained during partial or radical nephrectomy for sporadic RCC between January 2015 and January 2020. In all cases the submitted surgical specimens were handled according to the current guidelines of the Polish Society of Pathologists and complied with the recommendations of the ISUP and the WHO for specimen handling, sampling and reporting [46,47]. The tissue specimens were formalin-fixed and paraffin-embedded using a routine pathological tissue processing technique. All sections were then assessed for: tumor size, WHO/ISUP grade, presence and percentage of necrosis, sarcomatoid and rhabdoid differentiation, small vessel lymphovascular invasion, fibrous renal capsule invasion, perinephric fat invasion, renal sinus fat and vascular invasion of renal sinus vessels, and AJCC TNM pathologic stage of the primary tumor (pT). Follow-up data included: date of nephrectomy, survival status, date of death, and/or date of last follow-up. The hematoxylin- and eosin-stained slides from all cases were reviewed by two pathologists who assigned both a WHO/ISUP grade and eighth edition of the American Joint Committee on Cancer (AJCC) TNM pathological staging category [48].

### 4.2. Immunohistochemical Staining and Its Evaluation

For each case, a representative slide of the tumor and the corresponding paraffin block were selected. The immunohistochemical staining was performed in an automated immunostainer according to the manufacturer’s instructions (Table 8), and the same methodology which was used and described in detail in our previous study on non-clear cell RCCs [49].

The staining results were independently examined by two pathologists, who were completely blinded for medical and pathological data of patients. Semi-quantitative analysis was performed to evaluate the CD44, MMP-2, and MMP-9 expression. The modified Allred et al. method was used to evaluate both the intensity and the proportion of immunohistochemical staining [49,50]. The intensity scores ranged from negative to strong as follows: 0 = negative, 1 = weak, 2 = moderate, and 3 = strong. The proportion scores ranged from 0 to 5 and were categorized according to the positive tumor cells as follows: 0 = no staining, 1 = up to 1/100 positive cells, 2 = 1/100 to 1/10 positive cells, 3 = 1/10 to 1/3 positive cells, 4 = 1/3 to 2/3 positive cells, 5 = >2/3 positive cells. To calculate the total immunohistochemical score, the proportion and intensity scores were multiplied for each specimen (range from 0 to 15) [51]. Then, overall immunohistochemical scores were classified into three groups as follows: 0–5 as Group 1 (low expression), 6–10 as Group 2 (moderate expression) and 11–15 as Group 3 (high expression).

### 4.3. SNP Selection

Based on previous reports and bioinformatics database http://www.ncbi.nlm.nih.gov/snp/ (accessed on 5 January 2023) two SNP loci rs1642742 and rs779805 in the gene *VHL* were selected. In order to perform the necessary genetic tests, MagCore Genomic DNA FFPE One-Step kit of the MagCore isolation system (FFPE One-Step kit of the MagCore RBC Bioscience Corp New Taipei City Taiwan) was used; this material was later used for genetic studies (allelic discrimination of two polymorphisms of gene *VHL*). Following primers according to probe manufacturer TaqMan Thermo Fisher Scientific were used: *VLH* rs1642742 Context Sequence [VIC/FAM]GGACAGCTTGTATGTAAGGAGGTTT[A/G]TATAAGTAATTCAGTGGGAATTGCA and *VLH* rs779805 Context Sequence [VIC/FAM]GGCCTAGCCTCGCCTCCGTTACAAC[A/G]GCCTACGGTGCTGGAGGATCCTTCT.

### 4.4. DNA Extraction and Genotyping

DNA was extracted from all samples using MagCore Genomic DNA FFPE One-Step Kit for DNA isolation, and the genotype was identified via ROCHE LifeCycler 96 (ROCHE LifeCycler 96 Basel, Switzerland). Real-time fluorescent quantitative polymerase chain reaction (PCR) was done using TaqMan genotyping method (TaqMan Thermo Fisher Scientific Waltham, Massachusetts, United States) allelic discrimination to identify specific assays. One blank control (MQ) was placed for marking each time in each 96-well detection unit as the quality control. Two genotypes and their frequency for each tumor case were determined.

### 4.5. Statistical Analysis

The statistical software STATISTICA 13.1 (StatSoft Inc., Tulsa, OK, USA) and Microsoft Excel 2013 (Microsoft Corporation, Redmond, WA, USA) was used to perform all analysis. Quantitative data are presented as numbers, case percentage, mean and standard deviation (SD). The study population complies with the Hardy-Weinberg law, which assumes that in a theoretically described population, genes do not mutate from one allele to another, nor do new alleles arise, because by definition this changes their frequencies. The genotype distribution was compared between groups using the χ2 test with Yates correction. A *p*-value less than 0.05 was considered to be statistically significant.

## 5. Conclusions

To the best of our knowledge, this is the first research investigating the influence of rs1642742 and rs779805 in the *VHL* tumor suppressor gene on ccRCC risk and survival among Central European population. The results of our study can be used to indicate genetic diversity for the *VHL* gene in the local population.

In summary, no significant differences in genotype or allele frequencies of *VHL* gene polymorphisms were observed between ccRCC cases and control population in our study, and we have proposed this study to be negative with respect to ccRCC susceptibility. Additionally, we did not observe a significant association of each allele and genotype of these two SNPs with survival of patients with ccRCC. However, our results conclude that rs1642742 and rs779805 in the *VHL* tumor suppressor gene are associated with increased tumor size, which is the most important prognostic indicator of renal cancer. Moreover, our analysis showed that patients with genotype AA of rs1642742 have a trend towards higher likelihood of developing ccRCC within their lifetime, while allele G of rs779805 can have a preventive effect against the development of renal cancer in stage 1. Therefore, these SNPs in *VHL* may be useful as genetic tumor markers for the molecular diagnostics and for facilitating the development of new treatments for ccRCC patients. However, further comprehensive and detailed studies with larger sample sizes are warranted to determine the clinical relevance of the association of these two *VHL* SNPs with ccRCC.

## Figures and Tables

**Table 1 ijms-24-03778-t001:** Clinicopathological characteristics of patients.

Clinicopathological Features	
Number of cases [n (%)]	129 (100%)
Age, years [mean ± SD]	63 ± 10.7
Gender [n (%)]	
Female	53 (41.1%)
Male	76 (58.9%)
Type of operation [n (%)]	
Radical nephrectomy	78 (60.5%)
Partial nephrectomy (NSS)	51 (39.5%)
Tumor location [n (%)]	
Right kidney	71 (55.1%)
Left kidney	58 (44.9%)
Tumor size, cm (mean ± SD)	5.7 ± 3.4
Tumor stage [n (%)]	
pT1	75 (58.1%)
pT2	9 (6.9%)
pT3	44 (34.1%)
pT4	1 (0.9%)
WHO/ISUP grading [n (%)]	
G1	58 (44.9%)
G2	47 (36.4%)
G3	13 (10.2%)
G4	11 (8.5%)
Tumor necrosis area % (mean ± SD)	5.7 ± 1.8
Sarcomatoid area % (mean ± SD)	10.2 ± 2.86
Angioinvasion present [n (%)]	27 (20.9%)
Renal fibrous capsule invasion present [n (%)]	73 (56.6%)
Perinephric fat invasion present [n (%)]	21 (16.3%)
Renal sinus fat invasion present [n (%)]	23 (17.8%)
Renal sinus vascular invasion present [n (%)]	18 (14.0%)
Dead [n (%)]	48 (37.2%)

**Table 2 ijms-24-03778-t002:** CD44, MMP-2 and MMP-9 immunoreactivity of clear cell renal cell carcinoma cases.

Overall Immunohistochemical Score	CD44	MMP-2	MMP-9
Group 1 (low expression) [n (%)]	93 (72.1%)	127 (98.4%)	127 (98.4%)
Group 2 (moderate expression) [n (%)]	24 (18.6%)	2 (1.6%)	1 (0.8%)
Group 3 (high expression) [n (%)]	12 (9.3%)	0 (0%)	1 (0.8%)

**Table 3 ijms-24-03778-t003:** Genotype and allele frequencies of *VHL* polymorphisms among the ccRCC cases and control population.

**Genotypes of rs1642742**	**Cases Population**	**Database Control Population**	** *p* **
AA	46.6% (n = 60)	43.4% (n = 43)	NS
AG	17.8% (n = 23)	48.5% (n = 48)	NS
GG	35.6% (n = 46)	8.1% (n = 8)	NS
Alleles of rs1642742	Cases population	Database control population	
A	55% (n = 143)	68% (n = 134)	NS
G	45% (n = 115)	32% (n = 64)	NS
**Genotypes of rs779805**	**Cases population**	**Database control population**	
AA	47.2% (n = 61)	43.4% (n = 43)	NS
AG	45.7% (n = 59)	48.5% (n = 48)	NS
GG	7.1% (n = 9)	8.1% (n = 8)	NS
Alleles of rs779805	Cases population	Database control population	
A	71% (n = 181)	68% (n = 134)	NS
G	29% (n = 77)	32% (n = 64)	NS

NS—not statistically significant.

**Table 4 ijms-24-03778-t004:** Associations of the *VHL* polymorphisms with largest diameter of tumor.

**Genotypes of rs1642742**	**Tumor Size**					**Alleles of rs1642742**	**Tumor Size**				
	**≤4 cm**	**4–7 cm**	**7–10 cm**	**>10 cm**	** *p* **		**≤4 cm**	**4–7 cm**	**7–10 cm**	**>10 cm**	** *p* **
AA	24.04%	13.18%	6.20%	3.10%	NS	A	28.91%	14.50%	8.53%	3.49%	NS
AG	7.75%	4.65%	4.65%	0.77%	NS	G	12.40%	16.28%	7.75%	8.14%	NS
GG	8.53%	13.95%	5.43%	7.75%	NS						
*p*	NS	NS	NS	*p* = 0.05			NS	NS	NS	NS	
Genotypes of rs779805	Tumor size					Alleles of rs779805	Tumor size				
	≤4 cm	4–7 cm	7–10 cm	>10 cm			≤4 cm	4–7 cm	7–10 cm	>10 cm	
AA	24.04%	13.95%	6.20%	3.10%	NS	A	31.04%	21.31%	10.85%	6.97%	NS
AG	13.95%	14.73%	9.30%	7.75%	NS	G	9.30%	10.46%	5.42%	4.65%	NS
GG	2.34%	3.10%	0.77%	0.77%	NS						
*p*	NS	NS	*p* = 0.048	*p* = 0.05			NS	NS	NS	NS	

NS—not statistically significant.

**Table 5 ijms-24-03778-t005:** Associations of the *VHL* polymorphisms with pathologic tumor (T) stage of renal carcinoma.

**Genotypes of rs1642742**	**Tumor Stage**					**Alleles of rs1642742**	**Tumor Stage**				
	**1**	**2**	**3**	**4**	** *p* **		**1**	**2**	**3**	**4**	** *p* **
AA	28.68%	3.10%	14.73%	0%	*p* = 0.047	A	34.88%	3.48%	17.05%	0%	NS
AG	12.42%	0.77%	4.65%	0%	NS	G	23.25%	3.48%	17.05%	0.81%	NS
GG	17.05%	3.10%	14.73%	0.77%	NS						
*p*	NS	NS	NS	NS	NS		NS	NS	NS	NS	
Genotypes of rs779805	1	2	3	4		Alleles of rs779805	1	2	3	4	
AA	29.46%	3.10%	14.73%	0%	NS	A	41.86%	4.49%	23.83%	0.38%	NS
AG	24.82%	2.32%	17.05%	0.77%	NS	G	16.28%	1.93%	10.85%	0.38%	NS
GG	3.88%	1.55%	2.32%	0%	NS						
*p*	NS	NS	NS	NS			*p* = 0.05	NS	NS	NS	

NS—not statistically significant.

**Table 6 ijms-24-03778-t006:** Associations of the *VHL* polymorphisms with CD44, MMP-2 and MMP-9 expression levels.

	**Overall Immunohistochemical Score—CD44**				**Overall Immunohistochemical Score—CD44**		
Genotypes of rs1642742	Group 1	Group 2	Group 3	Alleles of rs1642742	Group 1	Group 2	Group 3
AA	34.88%	6.20%	5.42%	A	43.02%	6.97%	5.42%
AG	16.28%	1.55%	0%	G	29.84%	10.85%	3.87%
GG	21.70%	10%	3.87%				
*p*	NS	NS	NS		NS	NS	NS
Genotypes of rs779805	Group 1	Group 2	Group 3	Alleles of rs779805	Group 1	Group 2	Group 3
AA	35.65%	6.20%	5.42%	A	51.94%	11.24%	6.97%
AG	32.55%	10%	3.10%	G	20.93%	6.58%	2.32%
GG	4.65%	1.55%	0.77%				
*p*	NS	NS	NS		NS	NS	NS
	Overall immunohistochemical score—MMP-2				Overall immunohistochemical score—MMP-2		
Genotypes of rs164242	Group 1	Group 2	Group 3	Alleles of rs1642742	Group 1	Group 2	Group 3
AA	46.51%	0%	0%	A	56.58%	0%	0%
AG	17.83%	0%	0%	G	43.02%	1.55%	0%
GG	34.10%	1.55%	0%				
*p*	NS	NS	NS		NS	NS	NS
Genotypes of rs779805	Group 1	Group 2	Group 3	Alleles of rs779805	Group 1	Group 2	Group 3
AA	47.28%	0%	0%	A	69.40%	0.77%	0%
AG	44.18%	1.55%	0%	G	29.06%	0.77%	0%
GG	6.97%	0%	0%				
*p*	NS	NS	NS		NS	NS	NS
	Overall immunohistochemical score—MMP-9				Overall immunohistochemical score—MMP-9		
Genotypes of rs1642742	Group 1	Group 2	Group 3	Alleles of rs1642742	Group 1	Group 2	Group 3
AA	46.51%	0%	0%	A	55.81%	0.38%	0.38%
AG	17.05%	0.77%	0.77%	G	43.41%	0.38%	0.38%
GG	34.88%	0%	0%				
*p*	NS	NS	NS		NS	NS	NS
Genotypes of rs779805	Group 1	Group 2	Group 3	Alleles of rs779805	Group 1	Group 2	Group 3
AA	47.28%	0%	0%	A	69.40%	0.38%	0.38%
AG	44.18%	0.77%	0.77%	G	29.06%	0.38%	0.38%
GG	6.97%	0%	0%				
*p*	NS	NS	NS		NS	NS	NS

NS—not statistically significant.

**Table 7 ijms-24-03778-t007:** Polymorphisms of *VHL* and patients’ survival.

	**Deaths [n (%)]**				**Deaths [n (%)]**		
**Genotypes of rs1642742**	**Yes** **[n (%)]**	**No** **[n (%)]**	** *p* **	**Alleles of rs1642742**	**Yes** **[n (%)]**	**No** **[n (%)]**	** *p* **
AA	19 (14.73%)	41 (31.78%)	NS	A	22 (17.05%)	49 (37.98%)	NS
AG	6 (4.65%)	17 (13.18%)	NS	G	26 (20.16%)	32 (24.81%)	NS
GG	23 (17.83%)	23 (17.83%)	NS				
Genotypes of rs779805			*p*	Alleles of rs779805			*p*
AA	19 (14.73%)	42 (32.55%)	NS	A	30 (23.25%)	61 (47.30%)	NS
AG	22 (17.05%)	37 (28.67%)	NS	G	18 (13.95%)	20 (15.50%)	NS
GG	7 (5.44%)	2 (1.56%)	NS				

NS—not statistically significant.

**Table 8 ijms-24-03778-t008:** Details of the primary antibodies used for immunohistochemistry.

Antibody	Clone	Source	Dilution
CD44	Monoclonal	MRQ-13, Cell Marque, Rocklin, CA, USA	1:300
MMP-2	Monoclonal	CA-4001, Zeta Corporation, Arcadia, CA, USA	1:50
MMP-9	Monoclonal	EP127, Bio SB, Goleta, CA, USA	1:100

## Data Availability

Not Applicable.

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
