# Peer review of "Two Single Nucleotide Polymorphisms in the Von Hippel-Lindau Tumor Suppressor Gene in Patients with Clear Cell Renal Cell Carcinoma"

_ijms, 2023, doi:10.3390/ijms24043778_

Round 1

Reviewer 1 Report (Previous Reviewer 2)

The authors have addressed the main comments previously raised by the reviewer. Please include p-values in Table 3.

Author Response

Dear Reviewer,

Thank You for Your comment and suggestion. We tried to respond to them as best as we could to make our manuscript better quality.

Point 1: The authors have addressed the main comments previously raised by the reviewer. Please include p-values in Table 3.

Answer 1: Thank You very much for appreciating our effort to address all previous comments as best as possible. We have included p-values in Table 3, however they were not statistically significant.

Reviewer 2 Report (Previous Reviewer 1)

As I've reviewed the manuscript before and the authors responded to my comments, I do not have any other questions.

Author Response

Dear Reviewer,

Thank You for Your comment.   

Point 1: As I've reviewed the manuscript before and the authors responded to my comments, I do not have any other questions. 

Response 1: Thank You for appreciating our manuscript and the effort put into the revision of the study.

This manuscript is a resubmission of an earlier submission. The following is a list of the peer review reports and author responses from that submission.

Round 1

Reviewer 1 Report

The manuscript "Two single nucleotide polymorphisms in the von Hippel-Lindau tumor suppressor gene in patients with clear cell renal cell carcinoma" written by Chrabanska M, Szweda-Gandor N and Drozdozowska B. presents the analysis of two polymorphisms in VHL gene, in 129 patients with clear cell renal cell carcinoma. The results comprise analysis of patients' genotypes and allele frequencies of two polymorphisms, correlations' analysis with tumor size, stage and expression of MMP9, MMP2 and CD44 and patients' survival.

The manuscript is in general well-written. However, I have several comments. The Introduction could have more data on the biology and function of VHL. Also, the importance and the reason of CD44, MMP2 and MMP9 analysis could be explained.

The authors found correlation with tumor size, but they found that p is 0.05. Statistically, p should be less than 0.05 to be considered significant.

Other comments:

Abstract: several words are written in italic and in different font

line 17: full name of VHL should be written when mentioning for the first time

names of genes should be written in italics

line 73: characteristics were provided

line 76: experiment analyses expression of certain genes on the level of proteins. Immunohistochemical staining is just a method. According to that, also in Discussion and Results, there is no "immunoreactivity" of certain proteins, but protein expression. For comparison, normal renal tissue could be analyzed and protein expression commented.

line 87: there is no data on the origin of "control European population"

line 165: we could not find any...

line 172: investigated

line 177: studies cited above

some sentences in the Discussion are too long

line234: between ...and

line 279: were

Author Response

Dear Reviewer,

Thank You for all Your comments and suggestions. We tried to respond to them as best as we could to make our manuscript better quality.

  1. The manuscript is in general well-written. However, I have several comments. The Introduction could have more data on the biology and function of VHL. Also, the importance and the reason of CD44, MMP2 and MMP9 analysis could be explained.

Answer: We have added more information about biology and function of VHL gene in the “Introduction” section. We also explained the importance and the reason of CD44, MMP2 and MMP9 analysis in the “Introduction” section.

2. The authors found correlation with tumor size, but they found that p is 0.05. Statistically, p should be less than 0.05 to be considered significant.

Answer: We have changed in the text that the p-value is statistically significant if its value is less than 0.05. According to this, we also slightly changed the conclusion regarding tumor size.

Other comments:

1. Abstract: several words are written in italic and in different font                                                                                                                            Answer: In my version the whole text is in the same style, there are not any words appearing in italics or different font size. Maybe it is only due to the version of the Microsoft program in which the manuscript is opened.

2. Line 17: full name of VHL should be written when mentioning for the first time.                                                                                                               Answer: We have written the full name of VHL gene.

3. Names of genes should be written in italics.                                                    Answer: The names of all genes we wrote in italics.

4. Line 73: characteristics were provided                                                                  Answer: We corrected it.

5. Line 76: experiment analyses expression of certain genes on the level of proteins. Immunohistochemical staining is just a method. According to that, also in Discussion and Results, there is no "immunoreactivity" of certain proteins, but protein expression. For comparison, normal renal tissue could be analyzed and protein expression commented.                                                                       Answer: We have changed this formulation. Thank you for pointing this out.

6.Line 87: there is no data on the origin of "control European population".                                                                                                            Answer: All information about "control European population" came from the Ensembl database (http://www.ensembl.org/). We have included this information in the “Result” section (subsection 2.3.)           

7.Line 165: we could not find any...                                                                           Answer: We corrected it.

8.Line 172: investigated                                                                                      Answer: We corrected it.

9. Line 177: studies cited above.                                                                              Answer: We corrected it.

10. Some sentences in the Discussion are too long.                                         Answer: We have shortened the too long sentences in the “Discussion” section.

11. Line 234: between ... and                                                                                  Answer: We corrected it.

12. Line 279: were.                                                                                                   Answer: We corrected it.

Reviewer 2 Report

1) What is the ethnicity of the studied ccRCC cases? Where they all Polish? If so, why use European population, in general, as control group, and not Polish alone?

2) The number of the patients used i nthis study is rather small to infer enough statistical power and significant associations.

3) The authors mention that patients with genotype AA of rs1642742 have a higher likelihood of developing ccRCC within their lifetime, but the p value is marginal (p=0.047). I would be very conservative to highlight this as a stron finding.

4) Why not investigate the association between VHL polymorphisms with VHL immunoreactivity in ccRCC tumors?

5) Lines 165-166, the sentence needs to be rephrased as: "We couldn't find any significant difference in the VHL genotype and allele frequency between ccRCC patients and the control population..."

Minor:

1) There are several words appearing in italics and different font size, throughout the text. Why?

Author Response

Dear Reviewer,

Thank You for all Your comments and suggestions. We tried to respond to them as best as we could to make our manuscript better quality.

1) What is the ethnicity of the studied ccRCC cases? Where they all Polish? If so, why use European population, in general, as control group, and not Polish alone?

 Answer: Our population study consisted of patients from Upper Silesia in Poland (we have included this information in the “Materials and Methods” section – subsection 4.1.). All information about control population came from the Ensembl database (http://www.ensembl.org/). Due to the fact that in this database there is no examined Polish population, we chose for comparison the European population, which is the most similar to our Polish population among other possible selections in this database. 

2) The number of the patients used in this study is rather small to infer enough statistical power and significant associations.

Answer: We realize that our study was conducted on relatively small group of patients and may not have enough statistical power to show the significant associations. We stated it the limitation subsection at the end of the “Discussion” section. However, we consider this research relatively preliminary and plan further studies with larger number of patients.

3) The authors mention that patients with genotype AA of rs1642742 have a higher likelihood of developing ccRCC within their lifetime, but the p value is marginal (p=0.047). I would be very conservative to highlight this as a strong finding.

Answer: We realize that the p-value in this case is rather marginal, so we have changed the wording of this conclusion to be more careful

4) Why not investigate the association between VHL polymorphisms with VHL immunoreactivity in ccRCC tumors?

Answer: We did not investigate the association between VHL polymorphisms and VHL immunoreactivity in ccRCC tumors, because we had a problem with the purchase of an anti-VHL antibody. Moreover anti-VHL antibody is not signed by Dako and thus we would have to establish new protocols and make changes to the hardware system, which would significantly extend the workflow. Therefore we chose other antibodies such as CD44, MMP-2, and MMP-9, which have been investigated in many research as potential prognosticators for ccRCC. It is possible that in the future, if we manage to buy an anti-VHL antibody, we will investigate the association between VHL polymorphisms and VHL immunoreactivity.

5) Lines 165-166, the sentence needs to be rephrased as: "We couldn't find any significant difference in the VHL genotype and allele frequency between ccRCC patients and the control population..."

Answer: This sentence was rephrased – thank you for your watchfulness.

Minor:
1) There are several words appearing in italics and different font size, throughout the text. Why?

Answer: In my version the whole text is in the same style, there are not any words appearing in italics (except for the name of the genes)or different font size. Maybe it is only due to the version of the Microsoft program in which the manuscript is opened.  

Round 2

Reviewer 1 Report

The authors responded to my comments and I have no further remarks.

Reviewer 2 Report

The authors have tried to respond to the reviewer queries, but I am afraid that I am not satisfied with the extent of the revised version of the manuscript. Issues mainly deadling with the ethnicity of the control population and statistical significance, still remain. Therefore, I will have to reject the publication of this manuscript in the ijms.